# Bridging Functional Correctness and Runtime Efficiency Gaps in LLM-Based Code Translation

**Longhui Zhang** [1]  **Jiahao Wang** [1]  **Chenhao Hu** [1]  **Bingyu Liang** [1]  **Jing Li** [✉ 1]  **Min Zhang** [1]

## Abstract

While large language models (LLMs) have greatly advanced the functional correctness of automated code translation systems, the runtime efficiency of translated programs has received comparatively little attention. With the waning of Moore's law, runtime efficiency has become increasingly important for program quality, alongside functional correctness. Our preliminary study reveals that LLM-translated programs often run slower than human-written ones, and this issue cannot be remedied through prompt engineering alone. Therefore, our work proposes SWIFT-TRANS, a code translation framework comprising two key stages: (1) **Multi-Perspective Exploration**, where *MpTranslator* leverages parallel in-context learning (ICL) to generate diverse translation candidates; and (2) **Difference-Aware Selection**, where *DiffSelector* identifies the optimal candidate by explicitly comparing differences between translations. We further introduce *Hierarchical Guidance* for MpTranslator and *Ordinal Guidance* for DiffSelector, enabling LLMs to better adapt to these two core components. To support the evaluation of runtime efficiency in translated programs, we extend existing benchmarks, CodeNet and F2SBench, and introduce a new benchmark, SWIFTBENCH. Experimental results across all three benchmarks show that SWIFTTRANS achieves consistent improvements in both correctness and runtime efficiency.

## 1. Introduction

Code translation, the task of converting code from a source programming language (*e.g.,* C) to a target language (*e.g.,* Python), is vital in software engineering sce-

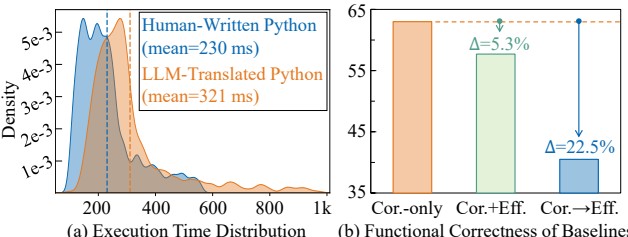

*Figure 1.* Challenges in runtime efficiency of LLM-translated code, shown on C-to-Python translation from F2SBench (Zhang et al., 2025b) with Qwen3-Next-80B (Qwen, 2025). (a) LLM-translated programs generally run slower than human-written ones. (b) This issue is hard to address, as prompt engineering strategies—such as prompts that additionally emphasize efficiency ("Corr.+Eff.") or employ post-hoc optimization ("Corr.→Eff.")—can improve efficiency but often reduce functional correctness relative to correctness-only prompts ("Corr.-only").

narios like legacy system migration and cross-platform development (Mossienko, 2003). The rise of large language models (LLMs) has introduced a new paradigm for code translation. Unlike earlier methods relying on handcrafted features (Zhong et al., 2010) or intricate deep architectures (Chen et al., 2018), LLMs can perform preliminary translation through simple prompt learning (Yan et al., 2023). This has attracted increasing research attention on enhancing the functional correctness of code translated by LLMs, and significant progress has been made (Jana et al., 2024; Zhang et al., 2025a; Ibrahimzada et al., 2025b).

Despite progress in **Functional Correctness** (Yin et al., 2024; Ibrahimzada et al., 2025a), **Runtime Efficiency** has received little attention in prior work. According to the ISO/IEC 25010 guidelines (ISO/IEC25010, 2011), programs with poor runtime efficiency may even be regarded as buggy. To address this gap, we conduct a preliminary investigation and present two key findings:

(1) *LLM-translated code typically exhibits lower efficiency than human-written code in the target language*, as shown in Fig. 1 (a). One major reason is that LLMs tend to replicate the logic and structure of the source code (Zhang et al., 2025b). Although such replication reduces the risk of errors, it also perpetuates any inefficient coding constructs present in the source code and neglects target language-specific optimizations, such as C pointers or Python's built-in functions.

---

[1]Harbin Institute of Technology, Shenzhen, China. Correspondence to: Jing Li <jingli.phd@hotmail.com>.

*Proceedings of the $43^{rd}$ International Conference on Machine Learning*, Seoul, South Korea. PMLR 306, 2026. Copyright 2026 by the author(s).

(2) *Ensuring both correctness and efficiency in translated code remains challenging*, as shown in Fig. 1 (b). Straightforward solutions, like complex prompts or post-hoc optimization modules, often improve efficiency at the cost of correctness due to increased complexity.

Our work introduces SWIFTTRANS, a code translation framework designed to ensure both correctness and runtime efficiency. SWIFTTRANS first employs a **Multi-Perspective Translator** (MpTranslator) to generate diverse translation candidates from the source code, and then applies a **Difference-Aware Selector** (DiffSelector) to identify the optimal one. MpTranslator draws on diverse, multi-scale demonstrations (*i.e.,* exemplar code translation pairs used as guidance), which improves translation quality and diversity compared to traditional repeated sampling (Brown et al., 2024). Through hierarchical guidance training, MpTranslator learns to produce outputs that range from conservative (correctness-first) to optimized (efficiency-aware) translations, enabling adaptation to tasks of varying complexity. Serving as a pairwise LLM-as-a-judge, DiffSelector performs fine-grained comparisons between translation candidates, considering both correctness and efficiency. It employs an efficient linear-time selection strategy, inspired by bubble sort, to evaluate all candidates. Finally, we introduce ordinal-guidance training to enhance DiffSelector's accuracy and robustness to candidate order.

Existing code translation benchmarks similarly focus primarily on evaluating functional correctness. Accordingly, we augment the widely used benchmarks CodeNet (Puri et al., 2021) and F2SBench (Zhang et al., 2025b) with manually curated, efficiency-critical test cases, together with explicit maximum runtime constraints on translated programs. Moreover, we introduce SWIFTBENCH, a new benchmark featuring source programs with inefficient code patterns, such as redundant computations or suboptimal algorithmic choices. This design evaluates whether translation models can eliminate inefficiencies in translated code. Extensive experiments on CodeNet, F2SBench, and SWIFTBENCH show that SWIFTTRANS consistently surpasses existing methods in both functional correctness and runtime efficiency.

Our key contributions are summarized as follows:

- We systematically identify and address runtime efficiency deficits in LLM-based code translation by proposing the SWIFTTRANS framework.

- We extend existing benchmarks and develop a new benchmark, SWIFTBENCH, to support the evaluation of both correctness and efficiency.

- Experiments across diverse benchmarks and programming languages show that our approach significantly improves the quality of translated code compared to various baselines.

## 2. Related Work

A number of studies have investigated how to improve the functional correctness of code generated by LLMs. These efforts can be broadly divided into two categories: training-free and training-based methods. Classic prompt learning strategies, such as RAG (Bhattarai et al., 2024a;b) and CoT (Yan et al., 2023), fall under training-free methods and have proven effective. Some studies leveraged compiler feedback to detect translation errors and guide LLM-based fixes (Yang et al., 2024; Pan et al., 2024; Ibrahimzada et al., 2025b). In contrast, training-based approaches employ well-designed training processes, which enable lightweight open-source LLMs to achieve translation performance comparable to proprietary models. For example, He et al. (2025) incorporated executability signals into training, substantially enhancing the executability of code. Zhang et al. (2025b) proposed a two-stage training paradigm combining supervised fine-tuning and preference learning. Jana et al. (2024); Wang et al. (2025) further optimized LLMs' code translation performance through reinforcement learning.

In addition to functional correctness, runtime efficiency is an important criterion for evaluating code quality (ISO/IEC25010, 2011). In the task of code generation, Gee et al. (2024) trained LLMs to produce efficient solutions to programming problems, thereby achieving end-to-end code generation with improved efficiency. Accelerating generated code via post-processing is another mainstream approach. For example, Shypula et al. (2024) investigated LLM-based strategies code acceleration using techniques such as RAG and CoT. Zhang et al. (2025c) further enhanced LLMs' optimization capabilities through curriculum learning. Although runtime efficiency has been increasingly recognized as an important metric for evaluating code generation models (Huang et al., 2024), to the best of our knowledge, existing research on code translation still focuses primarily on functional correctness. We argue that ensuring both functional correctness and runtime efficiency in translated code is crucial for applying code translation LLMs in practical software development.

## 3. Methodology

As shown in Fig. 2, given a source code snippet, our SWIFTTRANS first applies the *Multi-Perspective Exploration* to generate a diverse set of candidate translations, and then selects the optimal one through *Difference-Aware Selection*. In this process, LLMs provide critical support for SWIFTTRANS's two core components: *MpTranslator* and *DiffSelector*. We optimize LLMs specifically for these two components, enabling lightweight open-source LLMs (*e.g.,* Qwen2.5-3B) to match or even surpass the performance of powerful LLMs like GPT-5.

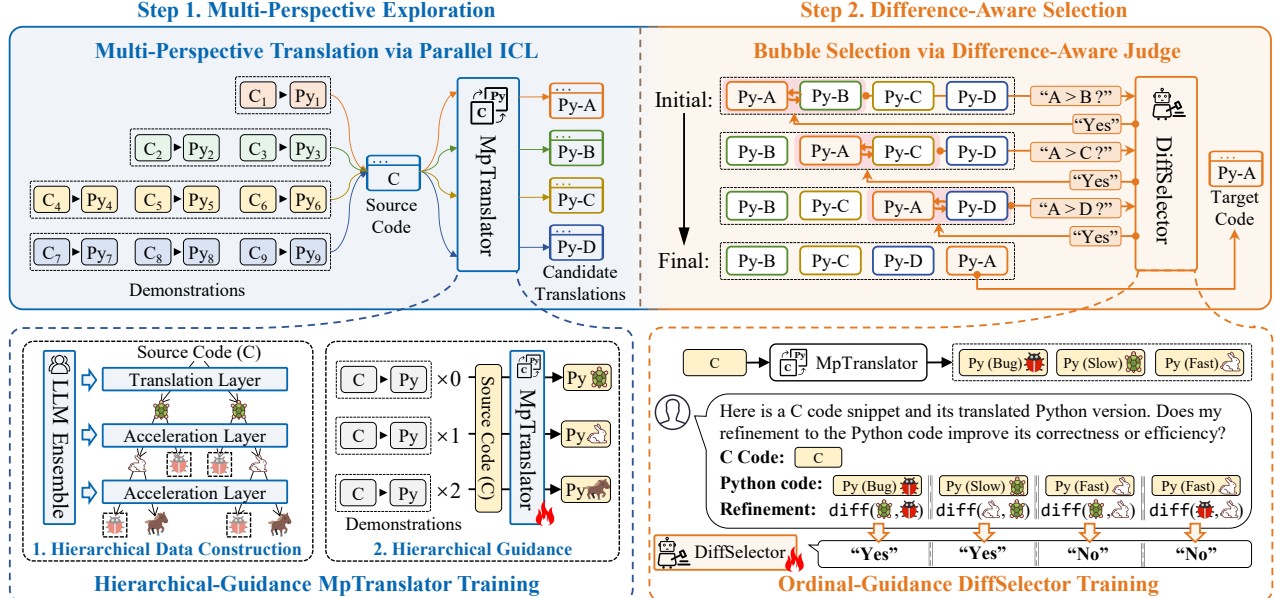

*Figure 2.* Overview of our SWIFTTRANS. Using C-to-Python translation as an example, *MpTranslator* first generates diverse candidates through parallel ICL, and *DiffSelector* applies a difference-aware judging strategy with bubble selection to identify the optimal one. We introduce *hierarchical* and *ordinal guidance* to train LLMs to better support MpTranslator and DiffSelector, respectively.

## 3.1. Multi-Perspective Exploration

This subsection first describes the multi-perspective translation mechanism of MpTranslator, which leverages parallel in-context learning (ICL) to generate diverse candidates. Next, it details the hierarchical guidance strategy used to optimize MpTranslator.

**Multi-Perspective Translation via Parallel ICL.** Traditional repeated sampling (Brown et al., 2024) generates multiple outputs from the same prompt. However, because the input context is fixed, the resulting outputs tend to stay within a narrow semantic space (Wang et al., 2024b).

To overcome this limitation, MpTranslator adopts parallel ICL to induce diverse translation behaviors. Specifically, given a source snippet $src$, it constructs $m$ demonstration sets, where each set contains a random number (0 to $K$) of demonstrations. Each demonstration is a source–target code translation pair sampled from a large library $\mathcal{C}$. This library $\mathcal{C}$ is derived from hierarchical guidance data, which will be discussed in the following section. MpTranslator generates one candidate translation per demonstration set, yielding $m$ candidates in parallel.

Compared to vanilla repeated sampling, MpTranslator offers two key advantages. First, demonstrations provide richer contextual signals, enabling higher-quality translations than zero-shot prompting. Second, by varying the context through multiple distinct demonstration sets—rather than repeatedly using the same prompt—MpTranslator produces genuinely diverse candidates.

**Hierarchical-Guidance MpTranslator Training.** To enhance the adaptability of lightweight, open-source LLMs to the MpTranslator, we employ the hierarchical guidance strategy grounded in instruction fine-tuning (IFT). Standard IFT optimizes LLMs via next-token prediction, improving their capacity to follow task-specific instructions. However, its direct application to MpTranslator faces two limitations: First, traditional IFT uses only the source code as input, while MpTranslator requires additional demonstrations as context during inference. This input inconsistency between training and inference can degrade model performance. Second, IFT learns from a single ground-truth response, which can lead to diversity collapse (Dang et al., 2025) in the model's outputs. To address these issues, we propose a hierarchical guidance training.

▶ *Hierarchical Data Construction.* We construct multi-level target code from source code collected on online platforms (*e.g.,* Codeforces). Lower levels correspond to functionally correct but slower implementations, while higher levels represent progressively optimized, faster versions.

Specifically, an ensemble of powerful LLMs (*e.g.,* DeepSeek-16B, Qwen3-30B) first generates initial translations focusing on functional correctness, with each LLM contributing one candidate. The ensemble then iteratively edits and accelerates these translations for up to $n$ rounds. A code compiler, leveraging online platform-provided test cases, filters out translations that are functionally incorrect or fail to achieve runtime improvement. Through this process, the ensemble contributes diverse strategies for transla-

tion and acceleration.

From each level, we randomly sample one code snippet, ensuring that each level exhibits at least a 10% speedup over the previous one. The source code $src$ and its most optimized translation $tgt^n$ are stored in the demonstration library $\mathcal{C}$. In addition, $src$ and its hierarchical translations $\{tgt^0, ..., tgt^n\}$ form our hierarchical training dataset, where $tgt^0$ is the initial functionally correct translation, and $tgt^{1...n}$ are increasingly optimized variants.

▶ *Hierarchical Guidance.* We use the constructed hierarchical data to train LLMs, yielding the final MpTranslator. First, for each source code $src$ and its target code $tgt^t$ at optimization level $t$, we randomly sample a subset $\mathcal{D}^t$ from the demonstration library $\mathcal{C}$, with the size of $\mathcal{D}^t$ set to $t$ to match the optimization level. For the base level $tgt^0$, which focuses solely on correctness, we set $D^0 = \emptyset$. We then train the model with demonstrations as context, with the loss defined as follows:

$$
\begin{aligned}
&\mathcal{L}_{\text{hg}}(src, \mathcal{D}^0, tgt^0, \ldots, \mathcal{D}^n, tgt^n) \\
&= -\frac{1}{n+1} \sum_t \sum_i \log p\left(tgt_i^t \mid \mathcal{D}^t, src, tgt_{<i}^t\right),
\end{aligned} \quad (1)
$$

where $tgt_i^t$ denotes the $i$-th token of $tgt^t$, and $tgt_{<i}^t$ represents the preceding token sequence.

This hierarchical guidance provides three key advantages: (1) Training with demonstrations as context ensures consistency between training and inference. (2) Learning from multiple translations per source mitigates the diversity collapse (Dang et al., 2025) inherent in standard IFT. (3) Linking the size $t$ of demonstration set $\mathcal{D}^t$ to the optimization level teaches the model to produce conservative translations under sparse context and increasingly efficient translations with richer context, thereby adapting flexibly to tasks of varying difficulty.

### 3.2. Difference-Aware Selection

This subsection first introduces the workflow of the DiffSelector component, which employs a difference-aware judge to evaluate translation quality and utilizes a bubble-selection strategy for optimal candidate selection. Next, it presents ordinal guidance, which optimizes DiffSelector to achieve greater accuracy and robustness.

**Bubble Selection via Difference-Aware Judge.** The LLM-as-a-judge strategy is commonly used to select the optimal candidate from multiple outputs generated by LLMs (Zheng et al., 2023). However, since translations originate from the same source code, their differences are often subtle, sometimes limited to only a few tokens. Distinguishing such minor variations is challenging for LLMs.

To address this, we introduce DiffSelector, a difference-aware selector designed to facilitate fine-grained discrimination among similar translations. DiffSelector adopts a pairwise comparison strategy, evaluating two translations at a time. Unlike conventional methods, it treats one translation as a modified version of the other, explicitly highlighting their differences to support more accurate judgments. As illustrated in Fig. 2, the `diff`$(tgt_1, tgt_2)$ operation shows the modifications from $tgt_1$ to $tgt_2$ in unified diff format, computed using GNU diff.

A straightforward use of DiffSelector is to compare every pair of candidate translations and select the best one, which requires $\mathcal{O}(n^2)$ comparisons for $n$ candidates. To improve efficiency, we draw inspiration from the bubble sort algorithm, in which elements are compared and swapped based on pairwise evaluations. Specifically, we utilize DiffSelector as the pairwise comparator and treat candidate translations as elements to be sorted by quality. As shown in Fig. 2, we first compare "Py-A" and "Py-B", retain the better one, and then compare it against the third candidate "Py-C". The process repeats sequentially until all candidates have been evaluated. In this way, DiffSelector identifies the best translation in a single pass with only $n-1$ comparisons, achieving $\mathcal{O}(n)$ complexity.

**Ordinal-Guidance DiffSelector Training.** We further enhance DiffSelector through ordinal guidance, which leverages the inherent ranking of translation quality: efficient and correct translations $\succ$ slower correct translations $\succ$ incorrect translations. Firstly, MpTranslator generates multiple candidate translations from source code $src$ collected on online platforms. Based on compiler feedback, we then select two target translations of different quality, denoted as $tgt^+$ and $tgt^-$. For example, $tgt^+$ is correct and efficient code, while $tgt^-$ is correct but less efficient. Given the source code $src$ and the two targets, we propose a bi-judge loss that trains the LLM to judge their relative quality bidirectionally, *i.e.,* whether $tgt^+$ constitutes an improvement over $tgt^-$ and vice versa. The loss function is defined as:

$$
\begin{aligned}
\mathcal{L}_{\text{og}}(src, tgt^+, tgt^-) = -\frac{1}{2}[&\log p(\text{"Yes"} \mid src, tgt^+ \succ tgt^-) \\
+ &\log p(\text{"No"} \mid src, tgt^- \succ tgt^+)],
\end{aligned} \quad (2)
$$

where "Yes" and "No" denote the ground-truth responses for the relative quality between $tgt^+$ and $tgt^-$. This bi-judge design mitigates sensitivity to candidate order (Zheng et al., 2023) in the prompt.

## 4. Experiments

### 4.1. Benchmark Construction

**Extension of Existing Benchmarks.** Current benchmarks, such as CodeNet (Puri et al., 2021) and F2SBench (Zhang et al., 2025b), primarily focus on func-

tional correctness but offer little support for efficiency evaluation, due to two main limitations: (1) The test cases are too simple to reveal runtime performance differences. For example, $\mathcal{O}(n^2)$ and $\mathcal{O}(n)$ implementations often show negligible runtime differences when $n = 1$. (2) The lack of baseline execution times prevents reliable efficiency evaluation. To address these limitations, we manually augment each sample in CodeNet and F2SBench with *ten efficiency-critical test cases* and *the maximum baseline execution time derived from conservative translations*. Annotation is performed by three independent teams, each consisting of 20 experienced software professionals. From the collected annotations, we select the ten most diverse and challenging test cases for each sample. For runtime evaluation, we annotate multiple conservative translations and adopt the slowest execution time among them as the reference.

**Construction of SWIFTBENCH.** Beyond extending existing benchmarks, we introduce a new benchmark, SWIFT-BENCH. Similar to CodeNet and F2SBench, SWIFTBENCH collects source code from online platforms and provides both efficiency-critical test cases and a baseline execution time of target code. Distinctively, each source program in SWIFTBENCH contains runtime efficiency issues, such as redundant computations or suboptimal algorithms. This design reflects real-world scenarios, where source code quality is often unpredictable. Consequently, SWIFTBENCH evaluates whether translation models can improve inefficient input programs. To mitigate benchmark leakage in LLM evaluation (Xu et al., 2024), SWIFTBENCH includes programming problems recently released on online platforms, specifically those released between June and October 2025. App. A provides additional details about the extended CodeNet, F2SBench, and SWIFTBENCH benchmarks.

### 4.2. Experimental Settings

#### 4.2.1. IMPLEMENTATION DETAILS.

In the multi-perspective translation via parallel ICL, we set the number of demonstration sets to $m = 10$, resulting in 10 candidate translations per source program, with each set containing up to $K = 3$ examples. For the hierarchical data construction, the LLM ensemble consists of DeepSeek-Coder-V2-16B, gpt-oss-20B, and Qwen3-Coder-30B, with the code acceleration depth $n$ fixed at 3. Our experiments cover translation among five programming languages: C, C++, Go, Java, and Python, yielding a total of 20 translation scenarios. Both the hierarchical guidance for MpTranslator and ordinal guidance for DiffSelector utilize approximately 15k training instances per scenario, consistent with the data scale in prior work (Zhang et al., 2025b). Both components are trained on the same set of open-source LLMs, such as Qwen2.5-3B, using full-parameter fine-tuning with a learning rate of 1e-5.

#### 4.2.2. METRIC DESIGN.

We evaluate translated code along two dimensions: **Computational Accuracy (CA)** and **Execution Time (ET)**. Computational Accuracy measures the proportion of translated programs that produce outputs identical to the source code across all inputs, following the standard metric used in prior work (Zhang et al., 2025b). Execution Time is defined as the average runtime of the translated code over all program inputs. For functionally incorrect translations, we use the baseline execution time from the benchmark as their runtime. To ensure reliable evaluation, we employ the Judge0 engine (Došilović & Mekterović, 2020), an online sandbox widely used for program execution testing (Waghjale et al., 2024). Each program, together with its inputs, is submitted to Judge0 and executed five times. The average runtime is then reported as the final result.

#### 4.2.3. BASELINES.

Our experiments include both training-free and training-based baselines. For the training-free baselines, we evaluate three prompt learning strategies on Qwen3-Next-80B (Qwen, 2025) and GPT-5: (1) **Correctness-Only**: prompts focusing solely on functional correctness. (2) **Correctness+Efficiency**: prompts emphasizing both correctness and runtime efficiency. (3) **Correctness→Efficiency**: a two-step prompting approach where the first step generates a correctness-oriented translation, which is then further optimized for efficiency. Detailed prompts for these training-free baselines are listed in the App. C. For the training-based baseline, we adopt **F2STrans** (Zhang et al., 2025b), which first applies IFT on weakly supervised data, followed by preference learning with high-quality data.

### 4.3. Main Results

We implement our SWIFTTRANS framework based on Qwen2.5-3B (Hui et al., 2024), Qwen2.5-7B and StarCoder-7B (Lozhkov et al., 2024) separately. Tab. 1 summarizes results on the extended CodeNet, F2SBench, and our newly constructed SWIFTBENCH across five programming languages (C, C++, Go, Java, Python), reporting averages from each source language to the other four targets. Sec. 4.5 presents additional benchmark results.

**Functional Correctness Evaluation.** Tab. 1 (I) shows that prompts aimed at improving efficiency often significantly reduce functional correctness, even for GPT-5. This is intuitive, as introducing efficiency-oriented constraints increases the complexity of code translation, amplifying the risk of logical errors. Although more powerful LLMs such as GPT-5 are more robust to this trade-off, their high inference costs hinder wide application. In contrast, With our SWIFTTRANS framework, Qwen2.5-3B achieves an aver-

*Table 1.* Functional correctness and runtime efficiency on CodeNet, F2SBench, and SWIFTBENCH. Results are averaged over translations from each source language to the other four (C, C++, Go, Java, and Python).

| Method | LLM | CodeNet | | | | | F2SBench | | | | | SWIFTBENCH (Ours) | | | | | Avg. |
|---|---|---|---|---|---|---|---|---|---|---|---|---|---|---|---|---|---|
| | | C | C++ | Go | Java | Py | C | C++ | Go | Java | Py | C | C++ | Go | Java | Py | |
| **(I) Functional Correctness Evaluation—Computational Accuracy (%) ↑** | | | | | | | | | | | | | | | | | |
| Cor.-Only | Qwen3-Next-80B | 79.3 | 81.5 | 71.2 | 77.3 | 80.9 | 69.7 | 61.0 | 64.8 | 75.2 | 50.4 | 75.1 | 75.8 | 84.2 | 81.6 | 71.4 | 73.3 |
| Cor.+Eff. | | 79.7 | 79.2 | 66.8 | 74.6 | 77.9 | 66.0 | 51.7 | 55.1 | 68.3 | 44.6 | 73.5 | 77.6 | 78.4 | 70.7 | 65.5 | 68.6 |
| Cor.→Eff. | | 68.9 | 72.9 | 69.5 | 69.3 | 70.1 | 50.0 | 43.9 | 48.0 | 50.9 | 35.5 | 61.7 | 60.5 | 67.6 | 62.3 | 58.1 | 59.3 |
| Cor.-Only | GPT-5 | 87.8 | 91.4 | 91.9 | 81.8 | 90.3 | 88.0 | 81.4 | 85.6 | 88.1 | **63.8** | 90.0 | 82.3 | 92.8 | 91.1 | 90.4 | 86.4 |
| Cor.+Eff. | | 82.9 | 88.5 | 89.1 | 81.2 | 80.5 | 79.9 | 72.9 | 78.5 | 84.1 | 50.1 | 83.3 | 75.0 | 88.3 | 79.8 | 83.6 | 79.8 |
| Cor.→Eff. | | 68.4 | 62.9 | 66.9 | 70.2 | 61.3 | 62.3 | 46.3 | 52.3 | 58.4 | 49.2 | 74.9 | 57.1 | 67.9 | 78.5 | 63.3 | 62.7 |
| F2STrans [ICML 2025] | Qwen2.5-3B | 86.4 | 89.8 | 85.6 | 86.5 | 83.6 | 84.8 | 73.0 | 79.4 | 85.2 | 44.8 | 86.6 | 86.5 | 90.9 | 87.2 | 79.9 | 82.0 |
| | Qwen2.5-7B | 91.0 | 91.4 | 86.8 | 88.5 | 91.1 | 85.6 | 75.6 | 82.2 | 86.7 | 49.6 | 87.8 | 88.6 | 92.8 | 88.4 | 83.1 | 84.6 |
| SWIFTTRANS (Ours) | Qwen2.5-3B | 91.8 | 92.7 | 89.7 | 93.4 | 94.0 | 87.5 | 80.5 | 81.4 | 88.5 | 59.9 | 89.1 | 84.3 | 91.7 | 91.3 | 88.1 | 86.9 |
| | Qwen2.5-7B | **93.6** | **95.0** | **96.1** | **94.9** | 94.6 | **91.2** | **82.7** | **86.9** | **90.3** | 62.1 | **93.1** | **92.3** | **96.5** | **93.1** | **91.5** | **90.2** |
| | StarCoder-7B | 92.3 | 93.8 | 95.4 | 94.3 | **94.8** | 89.7 | 82.0 | 85.4 | 88.7 | 61.5 | 92.4 | 91.5 | 95.4 | 92.1 | 91.0 | 89.4 |
| **(II) Runtime Efficiency Evaluation—Execution Time (ms) ↓** | | | | | | | | | | | | | | | | | |
| Cor.-Only | Qwen3-Next-80B | 514 | 685 | 363 | 174 | 315 | 1397 | 1164 | 523 | 257 | 356 | 1651 | 1729 | 983 | 856 | 682 | 776 |
| Cor.+Eff. | | 455 | 529 | 295 | 173 | 256 | 1274 | 814 | 419 | 222 | 339 | 1509 | 1538 | 823 | 774 | 586 | 667 |
| Cor.→Eff. | | 364 | 504 | 285 | 121 | 228 | 936 | 769 | 395 | 221 | 284 | 1186 | 1211 | 782 | 656 | 542 | 565 |
| Cor.-Only | GPT-5 | 391 | 435 | 355 | 161 | 187 | 766 | 801 | 373 | 223 | 288 | 1010 | 1071 | 783 | 594 | 484 | 528 |
| Cor.+Eff. | | 376 | 357 | 338 | 137 | 167 | 690 | 721 | 309 | 172 | 257 | 870 | 880 | 623 | 512 | 388 | 453 |
| Cor.→Eff. | | 322 | 329 | 328 | 126 | 143 | 645 | 675 | 278 | **131** | **197** | 747 | 753 | 582 | 394 | 344 | 399 |
| F2STrans [ICML 2025] | Qwen2.5-3B | 494 | 613 | 340 | 164 | 336 | 1239 | 1006 | 440 | 270 | 522 | 1532 | 1694 | 985 | 897 | 638 | 744 |
| | Qwen2.5-7B | 470 | 711 | 303 | 175 | 320 | 1228 | 1089 | 423 | 261 | 508 | 1518 | 1599 | 837 | 884 | 639 | 731 |
| SWIFTTRANS (Ours) | Qwen2.5-3B | 218 | 269 | 223 | 138 | 146 | 561 | 593 | 252 | 218 | 239 | 609 | 686 | 384 | 322 | 238 | 339 |
| | Qwen2.5-7B | **190** | **216** | **145** | **106** | **122** | **472** | **573** | **203** | 168 | 217 | **563** | **551** | **328** | **313** | **214** | **292** |
| | StarCoder-7B | 203 | 242 | 168 | 123 | 126 | 497 | 577 | 228 | 186 | 233 | 579 | 605 | 352 | 316 | 232 | 311 |

age CA 2.3% higher than F2STrans with Qwen2.5-7B, even though F2STrans leverages the stronger 7B model. Furthermore, applying SWIFTTRANS to Qwen2.5-7B outperforms GPT-5 by 3.8%. These results highlight both the potential of open-source LLMs for code translation and the effectiveness of SWIFTTRANS.

**Runtime Efficiency Evaluation.** From Tab. 1 (II), we can find that the "Correctness + Efficiency" and "Correctness → Efficiency" strategies do improve runtime efficiency, confirming that the target code translated by LLMs usually has significant room for efficiency improvement. However, these gains come at the expense of a decline in functional correctness, making these prompt engineering strategies suboptimal solutions. Moreover, scaling up F2STrans from 3B to 7B does not improve the runtime efficiency of translations. This stems from F2STrans's explicit emphasis on preserving the source code's logical structure (Zhang et al., 2025b), which mitigates errors but constrains runtime efficiency. In contrast, SWIFTTRANS employs multi-perspective exploration to generate diverse translations, fa-

cilitating the selection of candidates that better balance correctness and efficiency. For example, the execution time of code translated by Qwen2.5-7B-based SWIFTTRANS is comparable to that of code produced by GPT-5 under the "Correctness→Efficiency" strategy.

### 4.4. Analysis

We conduct detailed experiments to analyze our SWIFT-TRANS framework. Unless otherwise specified, the experiments are based on SWIFTTRANS with Qwen2.5-3B and evaluated on the SWIFTBENCH benchmark. Further discussions on SWIFTTRANS are provided in Sec. 4.5.

**Multi-Perspective Translation via Parallel ICL.** Fig. 3 illustrates the effect of the number of candidate translations $m$ and the number of demonstrations per perspective $k$. We evaluate $k$ under two settings: (i) fixed at 0, 1, 2, or 3, and (ii) randomly varying within [0,3].

As shown in Fig. 3, we observe that while ICL brings substantial benefits within our SWIFTTRANS framework, the

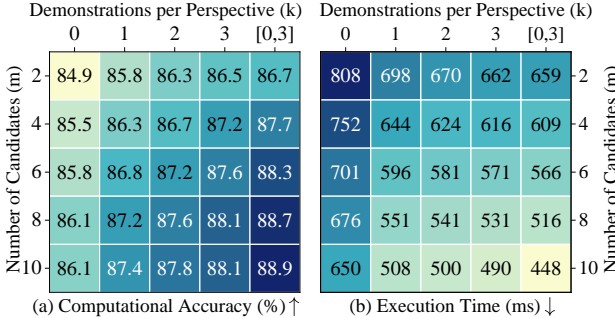

Figure 3. Effect of the number of demonstrations per perspective and translation candidates in multi-perspective translation.

gains begin to diminish as the number of demonstrations $k$ increases. For example, with $m = 10$ candidates, increasing $k$ from 0 to 1 improves CA by 1.3% and reduces ET by 142 ms, whereas increasing $k$ further from 1 to 3 provides only an additional 0.7% improvement in CA and 18 ms reduction in ET. Compared with using a fixed number of demonstrations, allowing $k$ to vary within $[0, 3]$ delivers larger gains. This is because translations generated under variable-$k$ settings are essentially an aggregation of translations from multiple fixed-$k$ settings, leading to a more diverse candidate pool. In addition, increasing the number of candidates $m$ consistently improves translation quality. This corroborates prior findings (Brown et al., 2024) that multiple generations from the same prompt can help push the boundaries of LLM performance.

**Hierarchical-Guidance MpTranslator Training.** We analyze two key aspects of the hierarchical guidance strategy: the number of acceleration layers $n$ used in hierarchical data construction and the loss function $\mathcal{L}_{hg}$ in Eq. 1.

▶ *The Number of Acceleration Layers $n$.* Fig. 4 (a) shows the results of SWIFTTRANS applying various numbers of acceleration layers. It can be observed that accelerating target code in the training data substantially mitigates efficiency issues in LLM-translated code, and additional acceleration layers further improve runtime efficiency. However, this comes at a slight cost to functional correctness—although the overall effect remains positive. For example, increasing $n$ from 0 to 4 significantly reduces ET by 425 ms, at the cost of a marginal decrease (0.4%) in CA.

▶ *The Loss Function $\mathcal{L}_{hg}$ of Hierarchical Guidance.* To analyze $\mathcal{L}_{hg}$, we define the following ablated variants of SWIFTTRANS: (1) "w/o $\mathcal{L}_{hg}$": candidate translations are generated directly by the base LLM, without hierarchical guidance training; (2) "w/o $\mathcal{D}^t$": demonstrations $\mathcal{D}^t$ are removed from $\mathcal{L}_{hg}$; (3) "$tgt^0$-only": only the correctness-first translation $tgt^0$ is used as the supervision signal; (4) "$tgt^n$-only": only the optimal translation $tgt^n$ is used as the supervision signal.

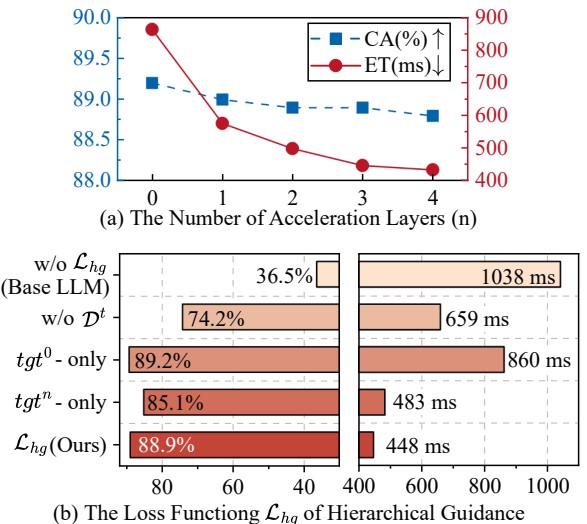

Figure 4. Analysis of the number of acceleration layers $n$ and the training loss function $\mathcal{L}_{hg}$ in hierarchical guidance.

Table 2. Comparison between all-pair and bubble selection. All-pair compares all candidate pairs, while bubble selection reduces comparisons to linear complexity.

| Method | CA (%) ↑ | ET (ms) ↓ | # Judge ↓ |
|---|---|---|---|
| All-Pair. | **89.1** | **439** | $\mathcal{O}(n^2)$ |
| Bubble. | 88.9 | 448 | $\mathcal{O}(n)$ |

Fig. 4 (b) shows that hierarchical guidance substantially improves the code translation performance of base LLMs on both CA and ET. The sharp performance drop in the "w/o $\mathcal{D}^t$" variant highlights the importance of ICL-based training for maintaining consistency between training and inference. Neither the "$tgt^0$-only" nor the "$tgt^n$-only" variant achieves balanced performance: The former fails to promote runtime optimization (ET = 860 ms), while the latter over-prioritizes efficiency at the expense of correctness (CA = 85.1%). In contrast, the full SWIFTTRANS maintains high correctness while improving efficiency, by integrating hierarchical guidance with multi-perspective generation to avoid the correctness–efficiency trade-off.

**Bubble Selection via Difference-Aware Judge.** Inspired by bubble sort, we introduce a bubble selection strategy to accelerate the candidate selection process of DiffSelector. We compare bubble selection with all-pair selection, which evaluates all candidate pairs before selecting the best one. As shown in Tab. 2, bubble selection matches the quality of all-pair selection while reducing comparisons from $\mathcal{O}(n^2)$ to $\mathcal{O}(n)$. Specifically, all-pair selection outperforms bubble selection by just 0.2% in CA and 9 ms in ET. Given this marginal performance difference and the significant reduction in the number of comparisons, bubble selection proves to be highly practical for efficient candidate evaluation.

*Table 3.* Ablation study on ordinal guidance for DiffSelector. The **Order Sensitivity (OS)** metric measures how sensitive the judge model is to the input order of translation pairs.

| Method | CA (%) ↑ | ET (ms) ↓ | OS (%) ↓ |
|---|---|---|---|
| **SWIFTTRANS** | **88.9** | **448** | **6.4** |
| w/o $\mathcal{L}_{og}$ | 86.1 | 609 | 64.2 |
| w/o diff | 87.3 | 519 | 27.5 |
| w/o Bi-Judge | 87.7 | 497 | 18.7 |

*Table 4.* Average memory usage and cyclomatic complexity of translated programs. Both F2STrans and our SWIFTTRANS use Qwen2.5-7B as the backbone.

| Method | Memory Usage (MB) ↓ | Cyclomatic Complexity ↓ |
|---|---|---|
| Qwen3-Next-80B | 27.6 | 6.5 |
| GPT-5 | 26.1 | 5.9 |
| F2STrans [ICML25] | 29.1 | 7.0 |
| **SWIFTTRANS** | **23.9** | **5.7** |

*Table 5.* Functional correctness and average inference time per sample across various code translation frameworks. For SWIFT-TRANS, the reported inference time accounts for generating all candidates and selecting the final translation.

| Method | Functional Correctness (%) ↑ | Inference Time (s) ↓ |
|---|---|---|
| Qwen3-Next-80B | 73.3 | 21.8 |
| GPT-5 | 86.4 | 121.3 |
| F2STrans [ICML25] | 84.6 | **5.1** |
| **SWIFTTRANS** | | |
| w/ 1-candidate | 87.1 | _5.6_ |
| w/ 5-candidate | _89.3_ | 7.6 |
| w/ 10-candidate | **90.2** | 9.8 |

and reduced cyclomatic complexity. These improvements stem from optimization strategies such as leveraging built-in library utilities and eliminating redundant logic, which naturally simplify control flow and reduce memory overhead. Tab. 6 shows the identified optimization types, with over half contributing to improvements in both metrics.

**Inference Efficiency of the SWIFTTRANS Framework.** Although generating multiple candidates and running the judge introduces additional inference cost, we highlight two points: (1) candidate generation in SWIFTTRANS is fully parallelizable, so the extra overhead remains limited, and (2) under the same inference budget, SWIFTTRANS still outperforms baselines.

Tab. 5 reports functional correctness and average per-sample inference time for various translation frameworks. Due to their larger sizes, Qwen3-Next-80B and GPT-5 incur much higher latency than F2STrans and SWIFTTRANS. With a single generated candidate, SWIFTTRANS has nearly the same inference time as F2STrans (a difference of only 0.2s) while achieving 2.5% higher correctness. Increasing the number of candidates from 1 to 10 roughly doubles the inference time but yields a 3.5% improvement in correctness.

**Ordinal-Guidance DiffSelector Training.** Ordinal guidance uses the loss $\mathcal{L}_{og}$ (Eq. 2) to compare translation quality bidirectionally. We conduct an ablation study on this loss function. To account for the fact that pairwise judging can be influenced by the order of translations in the prompt (Zheng et al., 2023), we further introduce the **Order Sensitivity (OS)** metric to evaluate this effect across judge model variants. OS quantifies the proportion of inconsistent judgments when the order of two translations is reversed. Lower OS values indicate greater robustness to input order.

Tab. 3 shows that all three ablated variants degrade CA, ET, and OS, confirming the effectiveness of ordinal guidance. Focusing on OS, we find that the base LLM exhibits high order sensitivity, with 64.2% of its judgments influenced by input order rather than translation quality, underscoring the limitations of off-the-shelf LLMs (Zheng et al., 2023). By explicitly incorporating diff information between translations and adopting the bi-judge training strategy, our ordinal guidance reduces this ratio to 6.4%. Importantly, the diff information contributes more than the bi-judge strategy, indicating that explicit difference information is crucial for distinguishing between highly similar translations.

### 4.5. Discussion

**Evaluation on Additional Code Quality Metrics.** Beyond functional correctness and runtime efficiency, Tab. 4 further evaluates translated code in terms of **Memory Usage** and **Cyclomatic Complexity**, the latter being a standard indicator of code maintainability. Although SWIFTTRANS primarily targets correctness and runtime performance, it additionally yields translations with lower memory consumption

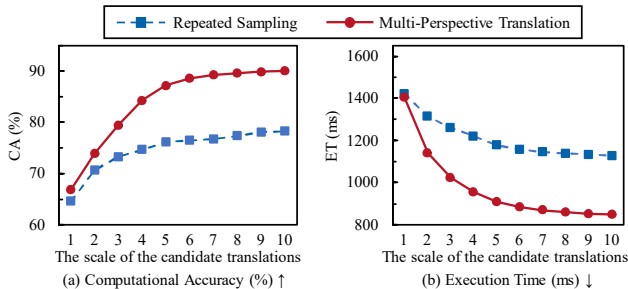

*Figure 5.* Comparison between the classic repeated sampling strategy and our multi-perspective translation strategy. In the experiment, Qwen3-Next-80B is used to generate multiple candidate Python translations for the C source code in the SWIFTBENCH benchmark, and the optimal one is selected.

**Comparison between Repeated Sampling and Multi-Perspective Translation.** We directly compare the classic repeated sampling approach with our multi-perspective translation strategy. We apply both translation strategies using Qwen3-Next-80B to translate the C-to-Python subset of SWIFTBENCH benchmark. Fig. 5 shows the pass@k results, where the best candidate translation is selected directly, without any judging process. It is evident that multi-perspective translation brings larger gains than repeated sampling. For instance, under multi-perspective translation, pass@10 improves by 23.2% over pass@1 on the CA metric, whereas repeated sampling only gains 13.7%. Furthermore, at pass@10, multi-perspective translation significantly outperforms repeated sampling on both CA and ET. These results confirm that our multi-perspective translation provides higher-quality candidates than simple repeated sampling.

*Table 6.* Distribution of optimization categories in 500 randomly sampled translations from Qwen2.5-3B-based SWIFTTRANS on SWIFTBENCH.

| Optimization Category | Percentage |
|---|---|
| Leveraging Language/Library Tools | 20.1% |
| Mathematical Simplification | 6.4% |
| Optimizing Algorithm Complexity | 13.4% |
| Removing Redundant Logic | **30.5%** |
| Upgrading Data Structures | 26.4% |
| Others | 3.2% |

**Categorization of Efficiency-Oriented Translation Optimizations.** We classify code optimization patterns into six categories: Leveraging Language/Library Tools, Mathematical Simplification, Optimizing Algorithm Complexity, Removing Redundant Logic, Upgrading Data Structures, and Others. To estimate the prevalence of each type, we randomly sample 500 translations produced by Qwen2.5-3B-based SWIFTTRANS on SWIFTBENCH and compare them with manually annotated SWIFTBENCH translations that are correct but inefficient. If multiple categories were involved in one example, we selected the one with the greatest impact. As shown in Tab. 6, most optimizations fall into three categories: Removing Redundant Logic, Upgrading Data Structures, and Leveraging Language/Library Tools.

*Table 7.* Functional correctness evaluation on real-world class-level and repository-level benchmarks.

| Method | Class level | Repository level | |
|---|---|---|---|
| | ClassEval-T | AlphaTrans | RepoTrans |
| Qwen3-Next-80B | 18.6 | 23.1 | 3.0 |
| GPT-4o | 25.7 | **29.1** | 4.0 |
| F2STrans [ICML 2025] | 21.6 | 16.6 | 0.0 |
| SWIFTTRANS | **28.4** | 27.5 | **7.3** |

**Generalization to Broader Real-World Benchmarks**
We further evaluate SWIFTTRANS on benchmarks covering broader real-world scenarios. These include the class-level ClassEval-T benchmark (Xue et al., 2025) and the repository-level AlphaTrans (Ibrahimzada et al., 2025a) and RepoTrans (Wang et al., 2024a) benchmarks. As shown in Tab. 7, SWIFTTRANS maintains strong performance across these benchmarks. The only exception is AlphaTrans, where GPT-4o achieves 1.6% higher functional correctness. We attribute this to the fact that source programs in AlphaTrans are very long, averaging over 5,000 tokens, a setting in which GPT-4o has a clear advantage over Qwen2.5-7B.

## 5. Conclusion

In this work, we proposed SWIFTTRANS, a novel code translation framework that ensures both functional correctness and runtime efficiency of translated programs. Given source code, SWIFTTRANS first uses MpTranslator to generate diverse candidates through a multi-perspective translation strategy, and then employs DiffSelector to select the correct and most efficient candidate after comparison. In addition, we introduced hierarchical guidance for MpTranslator and ordinal guidance for DiffSelector to better adapt LLMs to these two core components. To support runtime efficiency evaluation, we extended functionality-oriented benchmarks (CodeNet, F2SBench) and constructed a new benchmark, SWIFTBENCH. Extensive experiments across these three benchmarks demonstrate that SWIFTTRANS significantly improves the quality of LLM-based code translation.

## Impact Statement

This work aims to improve both correctness and runtime efficiency of LLM-based code translation, helping to reduce computational waste and enhance software reliability. The approach can benefit developer productivity and system performance. However, translated code should always be rigorously tested before deployment to avoid hidden bugs or security issues. The new benchmark also encourages more holistic evaluation in future research.

## Acknowledgements

This work was supported in part by National Natural Science Foundation of China (62476070), Shenzhen Science and Technology Program (JCYJ20241202123503005, GXWD20231128103232001, ZDSYS20230626091203008, KQTD20240729102154066), Department of Science and Technology of Guangdong (2024A1515011540) and National Key R&D Program of China (SQ2024YFE0200592).

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

## A. Benchmark Analysis

*Table 8.* Data statistics of CodeNet, F2SBench, and SWIFTBENCH.

| Benchmark | #Code | #Cases | Code Coverage | Branch Coverage | Date |
|---|---|---|---|---|---|
| CodeNet | $200 \times 5$ | 10 | 91% | 78% | Pre-2021 |
| F2SBench | $1000 \times 5$ | 10 | 86% | 75% | Mid-2024 |
| **SWIFTBENCH (Ours)** | $500 \times 5$ | 10 | 85% | 73% | Jun.–Oct. 2025 |

*Table 9.* Average execution time (ms) of conservative translations across benchmarks.

| Benchmark | {}→ C | {}→ C++ | {}→ Go | {}→ Java | {}→ Python |
|---|---|---|---|---|---|
| CodeNet | 241 | 358 | 402 | 820 | 594 |
| F2SBench | 296 | 431 | 714 | 1486 | 1290 |
| **SWIFTBENCH (Ours)** | 718 | 578 | 801 | 1814 | 1400 |

Tab. 8 presents the data statistics for the extended CodeNet, F2SBench, and our SWIFTBENCH. It can be seen that the test cases in these three benchmarks achieve average code coverage exceeding 85% and branch coverage above 73%, demonstrating the reliability of the annotated test cases. Additionally, Tab. 9 illustrates the average execution time of annotated conservative translations on these three benchmarks. It can be observed that the code samples in CodeNet tend to be relatively simple. In contrast, the source code in SWIFTBENCH is intentionally designed to include efficiency issues, resulting in slower execution times for the translated code. This highlights the challenging nature of the SWIFTBENCH benchmark.

## B. More Discussion

**Pair-wise vs. List-wise Selection Strategies.** While modern LLMs support extensive context windows that enable list-wise selection (evaluating all candidates in a single prompt), our experiments indicate that this approach consistently underperforms our pair-wise strategy. In the experiment, we first use the Qwen2.5-3B-based MpTranslator to generate 10 candidate translations for each source program in CodeNet, F2SBench, and SwiftBench. We then apply different selection variants to these candidates. As shown in Tab. 10, the pair-wise approach achieves superior accuracy and lower execution times across all model scales. We attribute this to the fact that list-wise selection often suffers from "lost-in-the-middle" effects and higher cognitive load (Liu et al., 2024), which can obscure subtle efficiency differences between candidates. In contrast, the pair-wise strategy allows the model to focus on fine-grained logic disparities, ensuring that the most performant implementation is identified even when candidates are highly similar. Consequently, we prioritize selection precision over the marginal throughput gains offered by list-wise processing.

*Table 10.* Comparison of pair-wise and list-wise selection strategies.

| Selection Strategy | Trained? | LLMs | Computational Accuracy ↑ | Execution Time ↓ |
|---|---|---|---|---|
| Pair-wise | No | Qwen2.5-3B | 84.6% | 482 ms |
| List-wise | No | Qwen2.5-3B | 83.8% | 524 ms |
| Pair-wise | Yes | Qwen2.5-3B | **86.9%** | **339 ms** |
| List-wise | Yes | Qwen2.5-3B | 86.3% | 357 ms |
| Pair-wise | No | Qwen3-Next-80B | 86.3% | 403 ms |
| List-wise | No | Qwen3-Next-80B | 86.0% | 420 ms |

**Plug-and-Play Gains with Inference-Only SWIFTTRANS.** A critical advantage of SWIFTTRANS lies in its flexibility as an inference-time framework, which does not require any additional training. As demonstrated in Tab. 11, SWIFTTRANS can be directly applied to off-the-shelf LLMs such as Qwen2.5-3B, Qwen2.5-7B, and Qwen3-Next-80B, yielding substantial improvements over their baseline performance on CodeNet. The results are compelling: even without fine-tuning, the inference-only version of SWIFTTRANS boosts computational accuracy by +15.1% for the 3B model, +8.4% for the 7B model, and +8.9% for the 80B model, while simultaneously reducing execution time by 51ms, 70ms, and 80ms, respectively. It suggests that the framework's value is not confined to scenarios where training data or resources are available; instead, it offers a powerful, plug-and-play solution for immediately boosting the quality of code translation from existing models.

*Table 11.* Performance of inference-only SWIFTTRANS on CodeNet across different backbone LLMs.

| LLM | Computational Accuracy ↑ (%) | Execution Time ↓ (ms) |
|---|---|---|
| Qwen2.5-3B | 44.8 | 416 |
| Qwen2.5-3B + SWIFTTRANS | 59.9 ($\Delta = +15.1$) | 365 ($\Delta = -51$) |
| Qwen2.5-7B | 65.4 | 402 |
| Qwen2.5-7B + SWIFTTRANS | 73.8 ($\Delta = +8.4$) | 332 ($\Delta = -70$) |
| Qwen3-Next-80B | 78.0 | 371 |
| Qwen3-Next-80B + SWIFTTRANS | 86.9 ($\Delta = +8.9$) | 291 ($\Delta = -80$) |

# C. Prompt Settings

---

**Multi-Perspective Translation.**

Translate the following {SOURCE_LANG} code into {TARGET_LANG} code, maintaining functionality, and optimizing for performance:
### {SOURCE_LANG} Code:
{SOURCE_CODE}
### {TARGET_LANG} Code:

---

**Difference-Aware Judge.**

Here is a {SOURCE_LANG} code snippet and its translated {TARGET_LANG} version. Does my refinement to the {TARGET_LANG} code improve its correctness or efficiency?
### {SOURCE_LANG} Code:
{SOURCE_CODE}
### {TARGET_LANG} Code:
{TARGET_CODE_1}
### Refinement:
diff({TARGET_CODE_1}, {TARGET_CODE_2})

---

**Translation Layer of Hierarchical Data Construction.**

Translate the {SOURCE_LANG} code to {TARGET_LANG} code.
### {SOURCE_LANG} Code:
{SOURCE_CODE}
### {TARGET_LANG} Code:

---

**Acceleration Layer of Hierarchical Data Construction.**

Below is a {SOURCE_LANG} code. Optimize the code and provide a more efficient version.
### {SOURCE_LANG} Code:
{SOURCE_CODE}
### Optimized Version:

---

**Correctness-Only Prompt.**

Translate the {SOURCE_LANG} code to {TARGET_LANG} code.
### {SOURCE_LANG} Code:
{SOURCE_CODE}
### {TARGET_LANG} Code:

---

**Correctness+Efficiency Prompt.**

Translate the following {SOURCE_LANG} code into {TARGET_LANG} code, maintaining functionality, and optimizing for performance:
### {SOURCE_LANG} Code:
{SOURCE_CODE}
### {TARGET_LANG} Code:

---

**Correctness→Efficiency Prompt.**

*Stage 1—Correctness-Only Prompt:*
Translate the {SOURCE_LANG} code to {TARGET_LANG} code.
### {SOURCE_LANG} Code:
{SOURCE_CODE}
### {TARGET_LANG} Code:
*Stage 2—Code Acceleration Prompt:*
Below is a {TARGET_LANG} code. Optimize the code and provide a more efficient version.
### {TARGET_LANG} Code:
{TARGET_CODE}
### Optimized Version:

## D. Case Study

Fig. 6 illustrates a case study of SWIFTTRANS, highlighting its advantages over traditional correctness-first models. Direct translation of the source code often carries over suboptimal logic from the original or overlooks optimizations specific to the target language. In contrast, SWIFTTRANS is designed to overcome these issues and produce translations that are both more efficient and more accurate.

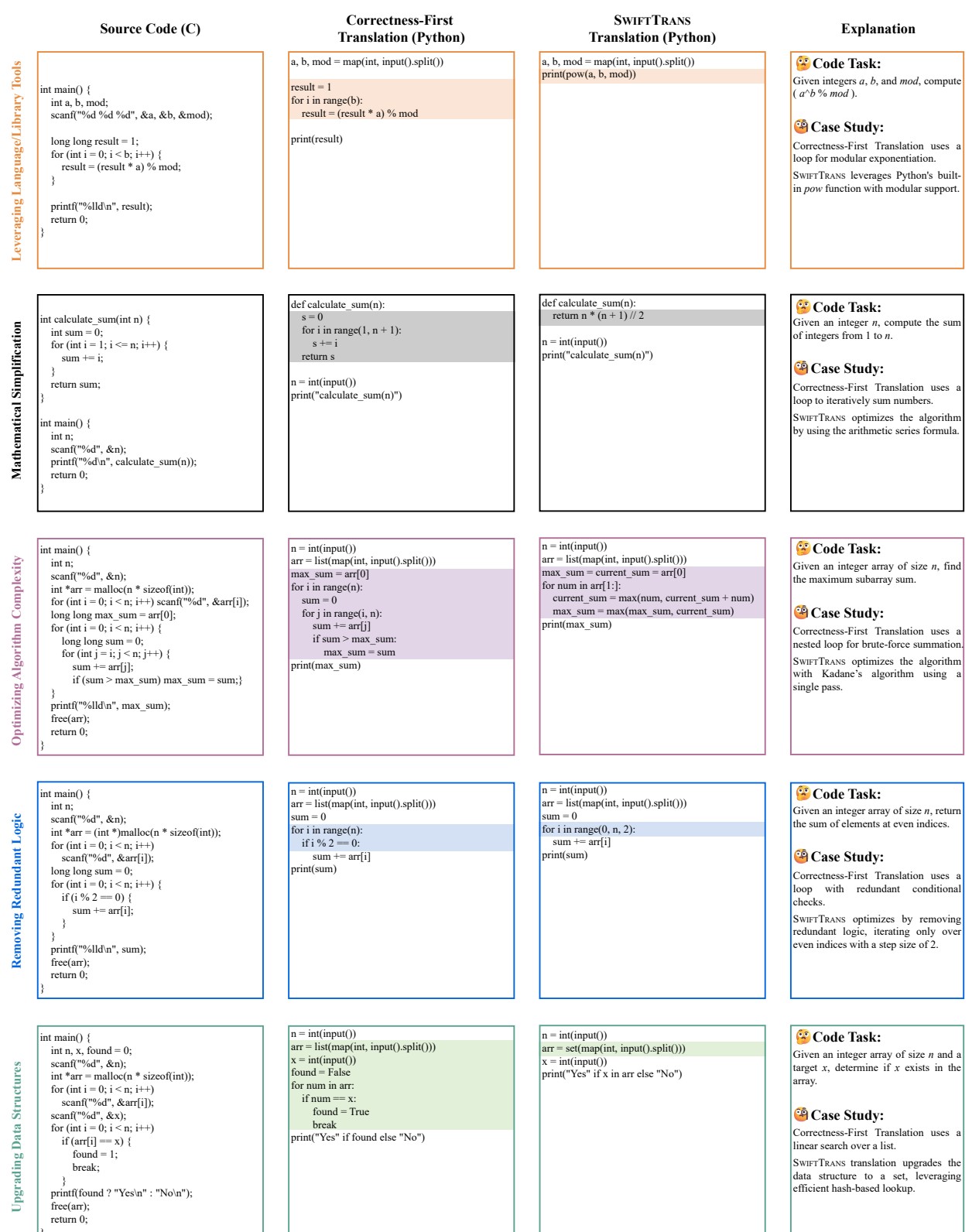

*Figure 6.* Case studies of SWIFTTRANS under different types of translation optimizations.

