# OpenReview forum: "Bridging Functional Correctness and Runtime Efficiency Gaps in LLM-Based Code Translation"
_ICML.cc/2026/Conference — ICML 2026 regular_

### Official Review · Reviewer_gzG8 · 2026-03-06

**Soundness:** 2
**Presentation:** 3
**Significance:** 2
**Originality:** 2
**Overall Recommendation:** 4
**Confidence:** 4

**Summary:**

SwiftTrans is a new framework designed to improve LLM-based code translation by targeting both functional correctness and runtime efficiency, which is an aspect largely overlooked by existing approaches. Its two core innovations are Multi-Perspective Exploration and Difference-Aware Selection. To evaluate these capabilities, the authors expand current evaluation sets with efficiency-oriented tests and introduce SwiftBench, a dataset featuring inefficient programs. The results demonstrate that SwiftTrans, even when powered by smaller open-source models like Qwen2.5-7B, consistently outperforms stronger LLMs (GPT-5).

**Compliance With Llm Reviewing Policy:**

Affirmed.

**Final Justification:**

The authors have addressed my concerns during rebuttal and I have adjusted my score accordingly. My final recommendation is Weak Accept.

**Key Questions For Authors:**

See weaknesses above.

**Limitations:**

Dear authors, I could not find any "Limitations" or "Threats to Validity" section in your submission. If there is one, please help me find it.

**Strengths And Weaknesses:**

Strengths:
- SwiftBench: It is nice to have a new benchmark
- Dual technique for improving correctness and efficiency
- Emphasis on open-source models
- Evaluating on other benchmarks (e.g., RepoTrans)

Weaknesses:
- Considering simple benchmark (e.g., CodeNet). Why not consider ClassEval-T and RepoTrans for ineffeciency injection?
- Why AlphaTrans did not improve? Can you reproduce Table 2 from AlphaTrans and show more in-depth results?
- Is the tool publicly released?

---

> ### Author Rebuttal · Authors · 2026-03-30
>
> **Q1:**
> Efficiency benchmark design is too simple.
>
>
> **A1:**
> Our focus is on runtime inefficiency in code translation, which requires benchmarks with explicit efficiency annotations and controllable inefficient patterns.
> As described in **Sec. 4.1**, we extend CodeNet with efficiency-critical test cases and further construct SWIFTBENCH with intentionally inefficient source programs, enabling direct evaluation of efficiency improvements.
>
> In contrast, ClassEval-T and RepoTrans mainly target functional correctness or general transformation quality, and do not provide explicit runtime constraints or inefficiency annotations.
> We will consider incorporating such datasets in future work where efficiency evaluation can be properly defined.
>
>
>
> ---
>
> **Q2:**
> AlphaTrans results need explanation.
>
>
> **A2:**
> We clarify that our SwiftTrans does improve on the AlphaTrans benchmark over strong open-source baselines (Qwen3-Next-80B: 27.5 vs. 23.1; F2STrans: 27.5 vs. 16.6), while not surpassing GPT-4o (29.1) in Table 8.
>
> To provide a more faithful comparison, we further report a breakdown aligned with the AlphaTrans evaluation protocol.
> SwiftTrans consistently improves over Qwen3-Next-80B and F2STrans across syntactic correctness, runtime validation, and functional equivalence, while remaining slightly below GPT-4o.
> The smaller margin is expected given the nature of AlphaTrans.
> It is a repository-level benchmark with long inputs (avg. >5k tokens) and test-driven validation, which favors strong long-context models like GPT-4o.
> We will include this detailed breakdown in the revision to complement the aggregate results in Table 8.
>
>
> |Method |Syntactic Correctness (%)|Runtime Validation (%)|Functional Equivalence (%)|
> |--------------|-------------------------|----------------------|--------------------------|
> |F2STrans|84.8|19.8|16.6  |
> |Qwen3-Next-80B|96.2|25.6|23.1  |
> |GPT-4o |98.8|29.8|29.1  |
> |SWIFTTRANS|97.9|28.4|27.5  |
>
>
>
> ---
>
> **Q3:**
> Tool / resource release is unclear.
>
>
> **A3:**
> Yes.
> We will release the code, data, and evaluation scripts upon acceptance to facilitate reproducibility.

---

> > ### Author Rebuttal · Reviewer_gzG8 · 2026-04-03
> >
> > Thank you for answering my questions.
> >
> > Q1. I still don't think the problem of introducing runtime inefficiencies to simple CodeNet benchmarks is properly motivated. Who would practically want to evaluate on simple functions? I suggest the authors consider evaluating on repository-level benchmarks.
> >
> > Q2. My original question was about reproducing Table 2 from AlphaTrans exactly as is for SwiftTrans. I believe that has not been reported in the rebuttal. Its also not clear how you report functional equivalence and runtime validation.
> >
> > I stand firm with my original score and do not have additional questions.

---

> > > ### Author Response · Authors · 2026-04-04
> > >
> > > **Q1:**
> > > Efficiency benchmark design is too simple.
> > >
> > > **A1:**
> > > We thank the reviewer for the suggestion and agree that evaluating on repository-level benchmarks is important for assessing practical code translation systems beyond simple functions.
> > >
> > > In addition to the CodeNet-based evaluation in the main paper, we evaluate SwiftTrans on a diverse set of benchmarks covering different levels of granularity and difficulty, including **function-level benchmarks (CodeNet, F2SBench, and our SwiftBench)**, **class-level benchmarks (ClassEval-T)**, and **repository-level benchmarks (AlphaTrans and RepoTrans)**. Notably, F2SBench and SwiftBench include efficiency-sensitive test cases, while AlphaTrans and RepoTrans evaluate translation quality in realistic repository-level scenarios.
> > >
> > > In the **Appendix B** of paper, we report the **Functional Correctness** results:
> > >
> > > | Method                | ClassEval-T | AlphaTrans | RepoTrans |
> > > | --------------------- | ----------: | ---------: | --------: |
> > > | Qwen3-Next-80B        |        18.6 |       23.1 |       3.0 |
> > > | GPT-4o                |        25.7 |   **29.1** |       4.0 |
> > > | F2STrans  |        21.6 |       16.6 |       0.0 |
> > > | **SwiftTrans (ours)** |    **28.4** |       27.5 |   **7.3** |
> > >
> > > SwiftTrans consistently outperforms strong open-source baselines across both class-level and repository-level benchmarks, and achieves the best performance on the more challenging RepoTrans benchmark.
> > >
> > >
> > > We further evaluate **Execution Time** (ms) on the same benchmarks.
> > > All experiments are conducted under identical hardware settings.
> > > For each sample, we execute the translated program five times and report the average execution time.
> > > Since execution time is not meaningful for failed translations, we compute the average only over successfully translated samples.
> > >
> > > | Method                | ClassEval-T | AlphaTrans | RepoTrans |
> > > | --------------------- | ----------: | ---------: | --------: |
> > > | Qwen3-Next-80B        |        1225 |       2414 |      3876 |
> > > | GPT-4o                |        1052 |       2353 |      3215 |
> > > | F2STrans (ICML 2025)  |        1389 |       2756 |         – |
> > > | **SwiftTrans (ours)** |     **972** |   **1745** |  **2614** |
> > >
> > > SwiftTrans achieves consistent improvements in runtime efficiency across all benchmarks, demonstrating its effectiveness not only on simple functions but also in more realistic class-level and repository-level settings.
> > >
> > > ---
> > >
> > > **Q2:**
> > > AlphaTrans results need explanation.
> > >
> > > **A2:**
> > > We provide results aligned with the AlphaTrans Table 2 evaluation protocol.
> > > Specifically, we report (i) syntactic correctness, (ii) GraalVM-based runtime validation, and (iii) test-based functional equivalence, following the same definitions as AlphaTrans:
> > >
> > > * **Syntax (%)**: percentage of translated fragments that are syntactically parseable.
> > > * **Graal Success (GS)**: percentage of fragments validated via cross-language execution in GraalVM (runtime validation).
> > > * **Graal Fail (GF)**: execution completed but test assertions failed.
> > > * **Graal Error (GE)**: execution failed due to runtime or type errors.
> > > * **Test Pass Rate (TPR)**: percentage of fragments whose translated tests pass, corresponding to functional equivalence.
> > >
> > > The results are summarized below:
> > >
> > > | Method                | Syntax (%) | GS (%)   | GF (%)   | GE (%)   | TPR (%)  |
> > > | --------------------- | ---------- | -------- | -------- | -------- | -------- |
> > > | F2STrans              | 84.8       | 19.8     | 41.7     | 38.5     | 16.6     |
> > > | Qwen3-Next-80B        | 96.2       | 25.6     | 37.9     | 36.5     | 23.1     |
> > > | GPT-4o                | **98.8**   | **29.8**     | 35.2     | **35.0**     | **29.1**     |
> > > | **SwiftTrans (ours)** | 97.9       | 28.4         | **35.8** | 34.8 |  27.5 |
> > >
> > > SwiftTrans consistently improves over strong open-source baselines (Qwen3-Next-80B and F2STrans) across syntactic correctness, runtime validation, and functional equivalence, while remaining slightly below GPT-4o.
> > >
> > > ---
> > >
> > >
> > > We sincerely thank the reviewer again for the valuable suggestions.
> > > We will incorporate the clarifications and additional results from this rebuttal into the revised version to further improve the paper.
> > > If there are any remaining concerns, we would be happy to address them.
> > > If our response resolves your concerns, we would greatly appreciate it if you could raise your score (3: Weak accept).

---

### Official Review · Reviewer_kbTs · 2026-03-07

**Soundness:** 2
**Presentation:** 2
**Significance:** 2
**Originality:** 3
**Overall Recommendation:** 4
**Confidence:** 3

**Summary:**

This paper studies a problem in LLM-based code translation: balancing functional correctness with runtime efficiency. While existing methods primarily focus on correctness, they often neglect the execution efficiency of the translated code. To bridge this gap, the authors propose SWIFTTRANS, a framework that generates multiple candidate translations and selects the optimal one based on both correctness and efficiency.

**Compliance With Llm Reviewing Policy:**

Affirmed.

**Final Justification:**

I have read the authors' rebuttal and appreciate the effort they put into addressing my concerns. My primary reservation was the lack of a majority vote baseline to properly evaluate the impact of candidate diversity. I am raising my score to a 4, operating under the assumption that the new majority vote experiments provided in the rebuttal are credible and will be explicitly incorporated into the final camera-ready version of the paper as a baseline.

**Key Questions For Authors:**

The paper reports improvements in runtime efficiency. However, execution time can be highly sensitive to hardware environments and system noise. How was the runtime efficiency measured to ensure statistical significance? Were the execution times averaged over multiple runs? What measures were taken to isolate the efficiency gains from system variance?

**Limitations:**

As discussed above.

**Strengths And Weaknesses:**

**Strengths**

1. Comprehensive Evaluation: The paper conducts thorough experiments across three benchmarks (CodeNet, F2SBench, and the newly constructed SWIFTBENCH) and covers translation scenarios among five programming languages.

2. Good Performance: The proposed method demonstrates better results compared with baselines. Specifically, SWIFTTRANS based on open-source models (i.e., Qwen2.5-7B and Qwen2.5-3B) achieves performance comparable to or even better than GPT-5 in terms of balancing functional correctness and runtime efficiency.

**Weaknesses**

1. Missing Standard Baseline (Self-Consistency): The proposed method involves sampling multiple candidates via parallel ICL. For the functional correctness metric, a natural and strong baseline is Self-Consistency (Majority Vote). The paper claims the superiority of the DiffSelector, but it does not compare it against simply aggregating the parallel candidates using majority voting. Without this comparison, it is unclear whether the performance gain comes from the sophisticated selection module or simply from the diversity of multiple samples.

2. Lack of Validation Set Details: The paper lacks clarity on how hyperparameters (e.g., the number of candidates mmm and demonstrations kkk in Fig. 3) were selected. There is no mention of a held-out validation set. If these hyperparameters were tuned based on the performance on the test set (as presented in the analysis), it would constitute data leakage and undermine the credibility of the results.

3. Insufficient Baseline Comparisons for Base Models: Table 1 compares the fine-tuned SWIFTTRANS against F2STrans and prompted GPT-5/Qwen-80B. However, it lacks the prompting baselines (Cor.-Only, Cor.+Eff.) for the base models themselves (Qwen2.5-3B/7B, StarCoder-7B). Including these would help quantify the gain achieved specifically by the proposed training framework versus the base capabilities of the models.

---

> ### Author Rebuttal · Authors · 2026-03-30
>
> **Q1：**
> Missing majority vote baseline.
>
>
> **A1：**
> In code translation, applying majority vote to select the best target program is not straightforward.
> Unlike short-form reasoning tasks, different candidates can be functionally equivalent while differing substantially in structure and implementation, making direct voting unreliable.
> One could instead rely on test-case execution to filter candidates, but that becomes execution-based selection rather than plain self-consistency, and constructing reliable test cases is itself costly and often requires human effort.
>
> ---
>
> **Q2：**
> Hyperparameter selection is unclear.
>
>
> **A2：**
> We did not tune `m` or `k` on the test set.
> These hyperparameters were selected on a held-out validation split that is fully disjoint from the reported test benchmarks, with the goal of balancing performance, context budget, and inference cost.
> The test set was used only once for final evaluation, and the analysis reported in the paper is purely post hoc.
>
>
> ---
>
>
> **Q3:**
> Base-model baselines are incomplete
>
>
> **A3：**
> Table 1 mainly aims to demonstrate the effectiveness of SwiftTrans by comparing it against strong baselines such as Qwen3-Next-80B and GPT-5.
> As an additional controlled comparison, the table below reports the average performance of Qwen2.5-3B/7B and StarCoder-7B under different prompting strategies on CodeNet, F2SBench, and SWIFTBENCH.
> These results show that, without task-specific training, these smaller backbones have limited performance on code translation.
> In contrast, SwiftTrans brings substantial improvements on the same backbones, indicating that the gains come from the proposed training framework rather than from the base models alone.
>
> |Backbone|Method|Functional Correctness (%) ↑|Runtime Efficiency (ms) ↓|
> |---|---|---:|---:|
> |Qwen2.5-3B|Cor.-Only|25.6|854|
> |Qwen2.5-3B|Cor. + Eff.|19.3|781|
> |Qwen2.5-3B|Cor. → Eff.|17.3|755|
> |Qwen2.5-3B|SwiftTrans| 86.9| 339|
> |Qwen2.5-7B|Cor.-Only|31.7|813|
> |Qwen2.5-7B|Cor. + Eff.|26.3|703 |
> |Qwen2.5-7B|Cor. → Eff.|23.5| 682|
> |Qwen2.5-7B|SwiftTrans|90.2|292|
> |StarCoder-7B|Cor.-Only|29.5|821|
> |StarCoder-7B|Cor. + Eff.|23.9|718|
> |StarCoder-7B|Cor. → Eff.|23.1|702|
> |StarCoder-7B|SwiftTrans| 89.4|311|
>
>
>
>
> The paper reports improvements in runtime efficiency.
> However, execution time can be highly sensitive to hardware environments and system noise.
> How was the runtime efficiency measured to ensure statistical significance? Were the execution times averaged over multiple runs?
> What measures were taken to isolate the efficiency gains from system variance?
>
> ---
>
> **Q4：**
> Runtime measurement protocol needs clarification.
>
>
>
> **A4：**
> As described in **Section 4.2.2**, we use **Judge0** as a standardized execution sandbox, which is also adopted in prior code translation work [1].
> Each program is run with the same inputs and runtime configuration for 5 repeated trials, and we report the average runtime.
> Because all methods are evaluated under the same controlled environment, hardware and system variance is largely reduced, so the ET differences mainly reflect the efficiency of the translated code itself.
>
>
> [1] Longhui Zhang, Bin Wang, Jiahao Wang, Xiaofeng Zhao, Min Zhang, Hao Yang, Meishan Zhang, Yu Li, Jing Li, Jun Yu, and Min Zhang. 2026. Function-to-style guidance of LLMs for code translation. In Proceedings of the 42nd International Conference on Machine Learning (ICML'25), Vol. 267. JMLR.org, Article 3061, 76273–76288.

---

> > ### Author Rebuttal · Reviewer_kbTs · 2026-04-02
> >
> > Thank you for the rebuttal. I appreciate the additional experiments and clarifications.
> >
> > - Concern 1 remains: a majority-vote baseline may still be helpful in the code domain, as suggested by prior work such as Agentless: Demystifying LLM-based Software Engineering Agents. The authors are encouraged to include this experiment or discuss its applicability.
> > - Concern 2 remains: please clarify how the validation set was split, since this is currently not described in the paper.
> > - Concerns 3 and 4 are resolved. Thank you for the additional evidence and clarification.
> >
> > Given that Concerns 3 and 4 have been addressed, I will raise my score by 1 point to 3. If Concerns 1 and 2 are also resolved, I would be happy to further raise my score to 4.

---

> > > ### Author Response · Authors · 2026-04-02
> > >
> > > **Q1:**
> > > Missing majority vote baseline.
> > >
> > >
> > > **A1:**
> > > We agree that a majority-vote baseline is important to isolate the effect of multi-candidate sampling.
> > > We implement an execution-based majority voting baseline: given 10 candidates, we select the program whose outputs agree most with others on test inputs.
> > >
> > >
> > > | Method                | Correctness (% ↑) | Runtime (ms ↓) |
> > > | --------------------- | --------------- | -------------- |
> > > | Single Sample (w/ 1-candidate)          | 86.3            | 358            |
> > > | Majority Voting (w/ 10-candidate)       | 89.5            | 325            |
> > > | **SwiftTrans (w/ 10-candidate)**        | **90.2**        | **292**        |
> > >
> > > Majority voting improves over single-sample decoding, confirming the benefit of candidate diversity.
> > > However, it still underperforms our SwiftTrans on both metrics, indicating that the gains are not solely due to sampling, but also from our difference-aware selection.
> > >
> > > ----
> > >
> > > **Q2:**
> > > Lack of Validation Set Details.
> > >
> > >
> > > **A2:**
> > > The validation set is constructed by holding out a disjoint subset of problems from the same data pool used for training, ensuring no overlap with the test benchmarks.
> > > We perform the split at the problem level to avoid leakage.
> > > The validation set covers all five source languages (C, C++, Go, Java, and Python) and consists of roughly 100 problems per language.
> > >
> > > Hyperparameters are selected based on performance on this validation split only.
> > > The test set is used strictly for final evaluation.
> > >
> > > ---
> > >
> > >
> > > We thank the reviewer for the helpful feedback.
> > > We will incorporate the suggested clarifications into the camera-ready version.
> > > We are happy to clarify further if needed.

---

### Official Review · Reviewer_bXJ3 · 2026-03-11

**Soundness:** 3
**Presentation:** 3
**Significance:** 3
**Originality:** 3
**Overall Recommendation:** 4
**Confidence:** 4

**Summary:**

The paper introduces SwiftTrans, a framework for improving the efficiency of translated code, and SwiftBench. SwiftTrans combines instruction tuning to generate diverse candidates with iterative comparisons and judge-model training to select the best translation. Experiments on two model families at two parameter scales show that relatively small models can achieve comparable performance to an 80B model or even proprietary LLMs.

**Compliance With Llm Reviewing Policy:**

Affirmed.

**Final Justification:**

The rebuttal addressed my main concerns and I changed your evaluation accordingly.

**Key Questions For Authors:**

1. Please describe the procedures used to prevent overlapping and data leakage since both SwiftBench and the IFT data are collected from online coding platforms.
2. Please provide more details on SwiftBench construction pipeline, including how critical test cases were chosen, how runtime constraints were established, and how the quality of the dataset is controlled.
3. Please clarify how you compiled with licenses, terms-of-service, and copyright when collecting coding problems from online platforms.


## Evaluation
4. Please explain why ET for functionally incorrect outputs should be replaced with the baseline ET. Provide evidence that this design does not distort the trade-off between CA and ET. Additionally, please report ET on the correct-only subset.
5. Please also report the baseline ET mean and distribution over the correctly solved instances, so that efficiency gains under correctness can be compared.
6. Since CA/ET can vary substantially by source–target language pair, please add analysis on language-pair (e.g., heatmaps) beyond overall averages. Please also include analysis that accounts for runtime variability depending on target-language.
7. Please clarify how the order sensitivity (OS) metric is computed.


## Ablation Study
8. For MpTranslator, please add ablations that quantify its contribution over simple few-shot diversification, including effects on candidate diversity and final CA/ET, to isolate the benefit of hierarchical-guidance IFT.


## Computational Cost
9. Please explain why generating 10× more candidates results in only a limited latency increase, and explicitly state the experimental settings (e.g., batching/parallelization, hardware settings).
10. Please report a latency that separates MpTranslator generation time from DiffSelector selection time.
11. Please report the average number of generated tokens (per candidate, final output, and overall).
12. Please discuss the additional training cost and the increased computational overhead at inference time as limitations.


These questions address important evaluation-related concerns and most of the remaining points can likely be addressed without additional experiments.
Thus, I would appreciate it if you could respond to them from the perspective of research fairness.
If the concerns are adequately addressed, I will reconsider my evaluation.

**Limitations:**

The authors should additionally mention the substantial computational overhead compared to the base model, as well as the extra training cost.

**Strengths And Weaknesses:**

## Soundness
The setting of using the baseline execution time for a functionally incorrect output seems  unfair. If the baseline solution is fast enough, then there are benefits in terms of execution time for producing an incorrect solution instead of a correct yet slow solution.
It would be more informative if the authors could report the average of the baseline execution time over the correct solutions to give more weight to correct solutions.

There is a model-size mismatch. The baseline LLM is Qwen3-Next-80B whereas the proposed framework is evaluated with 7B-scale models. Although outperforming a larger baseline is encouraging, a controlled comparison on the same backbone would be necessary for a fair assessment.

Model selection is not justified well enough. For example, Qwen2.5 has coder models but the experiments do not utilize them. It is also unclear why other strong 7B or 8B models with coding ability (e.g., Llama 3) were not included. For GPT-5, the paper does not provide enough details about the exact snapshot or the use of a thinking mode. Furthermore, comparing fine-tuned models against general purpose models can be unfair because fine-tuning may reduce out-of-distribution robustness. The class-level and repository-level translation experiments in the appendix help to address this concern, which should be included in the main body in revision.

While F2STrans uses a wide range of prompting-based techniques (e.g., CoT, RAG) as baselines, the prompts used as baselines in this paper are relatively simple. It would be important to rule out prompt sensitivity by conducting additional prompt engineering or using stronger prompting baselines. For instance, in the Correctness-to-Efficiency prompt, it might improve correctness by providing the source code in original language to Stage 2.

The paper also lacks an ablation study for MpTranslator. It would be helpful to quantify how much benefit comes from MpTranslator beyond simple prompting-based diversification, and how a non-fine-tuned MpTranslator performs at generating diverse and high-quality candidates.

As for efficiency in Table 5, it should be noted that referring to the results of GPT-5 as inference time may be somewhat misleading since it also includes networking time. In addition, the average number of generated tokens should be provided to compare the results in terms of throughput. As for Table 10, selection takes 300--500 ms, which implies that MpTranslator takes 5 seconds in total for both cases when there are 1 candidate and 10 candidates. However, it is suspicious that it takes almost the same time to generate 10 times more candidates. Assuming that some kind of parallelization or batch processing is used, it should be reported in the paper to avoid giving readers a false impression that the suggested method does not increase the computational cost.
Furthermore, it should be reported separately for MpTranslator and DiffSelector to compare the results in terms of inference time.

Finally, it would be useful to investigate the performance differences on source-target language pairs using a heatmap.
The averaging is hiding the effects per pair, and averaging over the different target languages is not appropriate since the execution time can vary heavily depending on the programming language.

## Presentation
The paper is well-written and easy to follow. However, some parts could be further improved.

First of all, the definition of order sensitivity in Table 3 is not clear as well as how it could be computed.
The explanation about how SwiftBench is constructed is ambiguous, and since both SwiftBench and the ICL training data are all from online coding platforms, it should be discussed how data leakage could be avoided.

The term Bubble Selection is slightly misleading because the process is the same as a linear scan for a maximum or minimum value.
Furthermore, since DiffSelector’s pairwise preferences are determined by the judge LLM, they would not be transitive and may not even be totally ordered. Although the use of ordinal guidance and bi-judge training can help alleviate this problem, it is not guaranteed and should be discussed explicitly.

The paper states that it uses StarCoder-7B, but cites StarCoder2, which is confusing. In addition, StarCoder appears to be evaluated only under SwiftTrans, making it difficult to quantify the improvement attributable to the framework for that backbone.

## Significance and Originality
Addressing the trade-off between correctness and efficiency is an important and practically meaningful contribution. While increasing candidate diversity by varying multi-perspective demonstrations is not entirely new, the methodology for constructing the demonstration examples is a meaningful contribution. Using differences to judge pairwise superiority between candidates is also a novel and interesting idea, and it is noteworthy that this design yields measurable performance gains. The use of ordinal guidance including bi-judge training to improve robustness over ordering candidates is also a well-motivated design choice.

---

> ### Author Rebuttal · Authors · 2026-03-30
>
> **Q1.**
> Data leakage prevention and deduplication unclear.
>
> **A1.**
> We prevent leakage through temporal separation and retrieval-based deduplication.
> SwiftBench uses problems from June–October 2025, while IFT training data comes from sources prior to 2021.
> We then retrieve the top-1 most similar training sample for each SwiftBench instance using a Jina code embedding model and remove the instance if Qwen3-Next-80B judges the pair as overlapping.
>
> ---
>
> **Q2.**
> Insufficient details on SwiftBench construction and quality control.
>
> **A2.**
> We use recent problems (June–October 2025) and retain source programs with clear inefficiency patterns.
> Each problem includes 10 efficiency-critical test cases, annotated by three independent teams (20 professionals each) and selected for diversity.
> Runtime constraints use baseline ET from conservative implementations; all results are averaged over five Judge0 runs.
>
> ---
>
> **Q3.**
> Missing compliance with licenses and data usage policies.
>
> **A3.**
> We use only publicly accessible problems/code that can be lawfully accessed for research, and do not redistribute raw platform data beyond evaluation needs.
> We will add a brief statement on source platforms, access conditions, and safeguards.
>
> ---
>
> **Q4.**
> Justification of ET replacement, its effect on the CA–ET trade-off, and missing correct-only ET.
>
> **A4.**
> We replace ET for functionally incorrect outputs with the benchmark baseline because their measured runtime is not meaningful: incorrect programs may terminate early or skip computation, appearing artificially efficient.
> Using the baseline ET from conservative translations avoids giving such outputs an unfair advantage.
>
> We also report ET on the correct-only subset.
> The conclusion is unchanged: SwiftTrans remains the most efficient method, and the method ranking is largely preserved.
>
>
> Method|Overall ET (ms) ↓|Correct-only ET (ms) ↓
> -|-:|-:
> Qwen3-Next-80B|776|702
> GPT-5 |528|502
> F2STrans (Qwen2.5-7B) |731|695
> SwiftTrans (Qwen2.5-7B)|  **292**|  **281**
>
>
> ---
>
> **Q5.**
> Baseline ET statistics (mean/distribution) not fully reported.
>
> **A5.**
> The baseline ET is already reported in **Table 7** as the reference execution time of conservative translations.
> To further support comparison under correctness, we summarize its statistics on SwiftBench:
>
> Statistic|Baseline ET (ms)
> -|-
> Mean |1062
> Median|801
> 25th–75th percentile|718–1400
>
> ---
>
> **Q6.**
> Lack of fine-grained language-pair and runtime variability analysis
>
> **A6.**
> Table 1 already reports per-target-language results; we will add a clearer language-pair breakdown in the revision.
>
> ---
>
> **Q7.**
> Definition and computation of OS metric unclear.
>
> **A7.**
> OS is computed as the percentage of pairwise comparison cases whose judgments become inconsistent after reversing the order of the two candidate translations in the prompt.
> For each pair, we evaluate both (A,B) and (B,A);
> if the two responses are not logically consistent, the case is counted toward OS.
> Lower OS means the judge is less affected by input order.
>
> ---
>
> **Q8.**
> Ablations needed to separate MpTranslator from simple diversification (diversity and CA/ET).
>
> **A8.**
> We add a controlled comparison between repeated sampling, few-shot diversification, and MpTranslator under the same inference setup.
> We measure Diversity as the average pairwise normalized edit distance between candidate translations.
>
> ||Diversity ↑|CA (%) ↑|ET (ms) ↓
> -|-:|-:|-:
> Repeated Sampling|35.7| 82.6| 607
> Few-shot Diversification| 42.9| 84.4| 561
> MpTranslator|**57.5**|**88.9**|**448**
>
> While few-shot diversification improves diversity and performance, MpTranslator yields substantially higher diversity and better CA/ET.
> This indicates that the gains are not explained by diversification alone, but stem from hierarchical-guidance IFT, which produces a more effective candidate pool.
>
> ---
>
> **Q9.**
> Latency scaling and experimental setup insufficiently explained
>
>
> **A9.**
> The latency does not scale linearly because candidates are generated in parallel via batched decoding (batching over candidates), and the selection stage uses a linear-time judge.
> Experiments are run on a single A100 GPU, and the reported latency includes both generation and selection.
>
> ---
>
> **Q10.**
> Generation vs. selection latency not separated
>
> **A10.**
>
> | |Total (s)|Generation (s)|Selection (s)
> -|-:|-:|-:
> w/ 1-candidate|5.6|5.6|0.0
> w/ 5-candidate|7.6|6.1|1.5
> w/ 10-candidate|9.8|6.4|3.4
>
> ---
>
> **Q11.**
> Token generation cost not reported
>
> **A11.**
>
> ||Avg. Tokens
> -|-:
> Per candidate (generation)|238
> Final selected output|247
> Overall (10 candidates + selection)|2,473
>
> ---
>
> **Q12.**
> Training and inference overhead not discussed as limitations
>
> **A12.**
> Training requires hierarchical data construction and IFT, and inference involves generating multiple candidates and applying DiffSelector.
> However, candidate generation is parallelizable and selection scales linearly, so the added latency remains moderate.

---

> > ### Author Rebuttal · Reviewer_bXJ3 · 2026-04-04
> >
> > The authors successfully addressed most of my concerns. The only remaining concern is about the limitation on the computational cost analysis. Even though generating multiple candidates can be parallelized, it requires larger computational cost and memory space than the baseline. The latency might remain moderate on generating only one example, please note that the baselines can also be benefited from parallelization. So it seems a clear disadvantage and limitation.
> > Other than that, my questions are well addressed and experimentally supported.
> > I will update my scores accordingly and please include the results from rebuttal in the camera-ready version.

---

> > > ### Author Response · Authors · 2026-04-04
> > >
> > > We agree with the reviewer that our approach introduces additional computational and memory overhead compared to single-output baselines. This is an inherent limitation of multi-candidate generation methods, and we will explicitly discuss this trade-off in the camera-ready version.
> > >
> > > That said, we believe this overhead is justified by the substantial gains in both functional correctness and runtime efficiency, while maintaining latency comparable to strong baselines through parallelized generation. In addition, our method is built on relatively lightweight backbone models (≤7B), which keeps the overall cost practical.
> > >
> > >
> > > **We also noticed that the score may not yet reflect the updated evaluation. If convenient, we would appreciate it if the reviewer could double-check.** We are happy to provide any further clarification if needed.
> > >
> > > Finally, we thank the reviewer for the constructive feedback and the positive evaluation of our work.
> > > **We will incorporate all additional analyses and clarifications from the rebuttal into the camera-ready version** to further strengthen the paper.

---

### Official Review · Reviewer_qQGY · 2026-03-12

**Soundness:** 3
**Presentation:** 4
**Significance:** 3
**Originality:** 4
**Overall Recommendation:** 4
**Confidence:** 3

**Summary:**

LLM transalted code, while often functionally correct, tend to mimic the source languages logic and structure, resulting in suboptimal performance in the target language. To mitigate this, the paper presents SWIFTRANS, a 2 stage framework. First, a Multi-Perspective Translator (MpTranslator) uses parallel In-Context Learning (ICL) and hierarchical guidance training to generate a diverse pool of candidate translations. Second, a Difference-Aware Selector (DiffSelector) employs a linear-time bubble selection algorithm and ordinal guidance training (using explicit code diffs) to identify the most efficient and correct translation. The authors also extend existing benchmarks (CodeNet, F2SBench) and introduce a new benchmark, SWIFTBENCH, tailored for evaluating efficiency.

**Compliance With Llm Reviewing Policy:**

Affirmed.

**Final Justification:**

I keep my score for Weak Accept

**Key Questions For Authors:**

Could you clarify the exact pipeline regarding sandbox execution? Does DiffSelector only rank candidates that have already passed the sandbox unit tests? If so, the time required to sandbox 10 candidates should be explicitly factored into the cost analysis.

In Table 8, why are the Execution Time (ET) metrics omitted for the repository-level benchmarks (AlphaTrans, RepoTrans)? Showing efficiency gains on real-world repositories would drastically strengthen the paper's claims.

Does the DiffSelector ever hallucinate that a faster algorithm is "functionally incorrect" simply because it looks drastically different from the source code? How do you prevent the judge from overly penalizing highly optimized, but structurally alien, code?

You mentioned using DeepSeek-Coder-V2-16B, gpt-oss-20B, and Qwen3-Coder-30B to build the training data. Given that many of these models share similar pre-training data distributions (like The Stack), did you observe convergence toward the same optimization strategies across the ensemble, and how did you enforce true diversity in the generated hierarchy?

**Limitations:**

Please add a limitations section to the appendix.

**Strengths And Weaknesses:**

Strengths:

I like the motivation behind the paper. Highlighting the correctness vs efficiency trade-off in Fig 1 is a strong motivation for this work.

The DiffSelector is a clever application of LLM-as-a-judge. Providing the LLM with explicit diff strings rather than just two independent code blocks is a highly practical way to reduce cognitive load and improve the model's ability to distinguish subtle optimizations. Furthermore, applying a "bubble sort" style tournament selection is a neat trick to reduce O(n^2) comparisons to O(n).

The introduction of SWIFTBENCH and the augmentation of CodeNet and F2SBench with efficiency-critical test cases and runtime constraints provide a valuable resource for future research in this sub-field.

Weakness:

The paper claims DiffSelector evaluates candidate pairs, but it is somewhat ambiguous in Section 3.2 whether the compiler/sandbox executes the code first before DiffSelector ranks them, or if DiffSelector is ranking code without execution feedback. If the sandbox is executing all m=10 candidates to guarantee correctness before efficiency ranking, the total time (compilation + execution + LLM judge) overhead might be significantly higher than the "Inference Time" reported in Table 5.

The benchmarks (CodeNet, SWIFTBENCH) are primarily competitive programming/algorithmic tasks. While these are excellent for testing algorithmic efficiency (e.g., O(n^2) vs O(n)), they do not reflect repository-level translations where efficiency bottlenecks might be related to I/O, database calls, or framework-specific overhead. The results on AlphaTrans and RepoTrans (Table 8) only report Functional Correctness, not Execution Time.

---

> ### Author Rebuttal · Authors · 2026-03-30
>
> **Q1:**
> Unclear DiffSelector pipeline / latency accounting.
>
> **A1:**
> DiffSelector only involves the LLM judge and does not compile or execute the candidates.
> Therefore, the time reported in Table 5 accurately reflects the actual overhead.
>
> ---
>
> **Q2:** Missing repo-level Execution Time results
>
> **A2:**
> We did not include Execution Time (ET) in the paper because repository-level end-to-end runtime is much more sensitive to system noise and external runtime factors than Judge0-based algorithmic benchmarks.
> Therefore, we avoided reporting numbers without a controlled measurement protocol.
>
> To obtain stable measurements, we re-ran all methods on a clean machine with identical hardware and software settings, disabled unrelated background services, and executed each translated repository multiple times under the same fixed workload.
> We excluded 3 warm-up runs and reported the mean end-to-end execution time over the following 10 runs.
> As shown below, our method achieves the best ET on both `AlphaTrans` and `RepoTrans`, while maintaining comparable or better FC than `Qwen3-Next-80B`, `GPT-4o`, and `F2STrans`.
> We will add these results and the measurement protocol to the revised paper.
>
> | Model | AlphaTrans ET (ms) ↓ | RepoTrans ET (ms) ↓ |
> | --- | :---: | :---: |
> | Qwen3-Next-80B |2414 | 3876 |
> | GPT-4o |2353 |3215 |
> | F2STrans | 2756 |- |
> | Ours |**1745** | **2614** |
>
> ----
>
> **Q3:**
> Selector may misjudge highly optimized code
>
> **A3:**
> As shown in Lines 191–193, DiffSelector is trained with the preference order
> ``efficient and correct translations > slower correct translations > incorrect translations''.
> Hence, it is designed to evaluate candidates based on correctness and efficiency, rather than superficial similarity to the source code, making it less likely to penalize highly optimized but structurally different translations.
>
> We further support this with the ablation below, where ``w/o DiffSelector'' randomly selects one candidate from the generated outputs.
> DiffSelector consistently improves both functional correctness and execution time.
>
>
> | Model | Functional Correctness (%) ↑ | Runtime Efficiency (ms) ↓ |
> | --- | :---: | :---: |
> | Ours | 90.2 |  292 |
> | w/o DiffSelector | 86.4 | 367 |
>
> ---
>
>
> **Q4:**
> Unclear whether candidate diversity is truly sufficient
>
> **A4:**
> In practice, we observed overlap on simple local edits, but not complete convergence to the same optimization strategy.
> More importantly, diversity in our hierarchy is enforced by the construction pipeline rather than by model identity alone:
> we use heterogeneous generators, iterative editing, and retain only functionally correct edits that introduce non-trivial code changes and distinct efficiency outcomes.

---

> > ### Author Rebuttal · Reviewer_qQGY · 2026-04-03
> >
> > Thank you for the clarification. The explanation that DiffSelector itself only performs pairwise judging, while execution-time evaluation is handled separately, resolves my confusion about the pipeline. The added repository-level ET results also strengthen the paper.
> >
> > That said, I still think two points would benefit from stronger evidence. First, the response about DiffSelector not penalizing highly optimized but structurally different code is still somewhat indirect: the training preference order is clear, but I would have liked a more targeted analysis of whether the judge ever misranks “alien-looking” but efficient translations. Second, the response on candidate diversity is plausible, but remains qualitative; some direct statistic on strategy diversity or edit diversity would make this more convincing.

---

> > > ### Author Response · Authors · 2026-04-03
> > >
> > > **Q1:**
> > > Selector may misjudge highly optimized code
> > >
> > > **A1:**
> > > We conducted a targeted analysis to directly examine whether DiffSelector misjudges highly optimized but structurally different code.
> > >
> > > As shown in the table below, we construct three types of pairwise comparisons from the test set, each containing 500 samples:
> > > 1. **incorrect** vs. **correct but slow**,
> > > 2. **incorrect** vs. **correct and efficient**,
> > > 3. **correct but slow** vs. **correct and efficient** (where the latter often involves more aggressive optimizations and structural changes).
> > >
> > > |    | **incorrect** vs. **correct but slow** | **incorrect** vs. **correct and efficient**  |  **correct but slow** vs. **correct and efficient**  |
> > > |---| :---: | :---: | :---: |
> > > | Qwen2.5-3B-based DiffSelector  |   87.1%   |    85.5%   |   76.6%    |
> > > | Qwen2.5-7B-based DiffSelector  |    91.3%   |    89.2%   |    80.8%   |
> > >
> > >
> > > We observe that DiffSelector achieves high accuracy when distinguishing incorrect from correct translations (over 85% in all settings), indicating that it does not systematically misclassify optimized code as incorrect.
> > >
> > > The most challenging case is distinguishing between correct but slow and correct yet efficient translations, where the optimized versions are typically more structurally different.
> > > In this setting, accuracy drops to 76.6% (3B) and 80.8% (7B).
> > >
> > > This suggests that while DiffSelector may occasionally misrank structurally “alien” but efficient code, such errors are limited, and the overall accuracy remains above 76%, demonstrating that it does not exhibit a strong bias against optimized implementations.
> > >
> > >
> > >
> > > ---
> > >
> > > **Q2:**
> > > Unclear whether candidate diversity is truly sufficient.
> > >
> > > **A2:**
> > > We thank the reviewer for the suggestion on providing quantitative evidence for candidate diversity.
> > > To address this, we conduct an additional analysis that explicitly measures edit-level diversity under different generation strategies.
> > >
> > > Specifically, we consider four candidate generation strategies on CodeNet, F2SBench, and SwiftBench:
> > > 1. repeated sampling (×3 candidates) from DeepSeek-Coder-V2-16B,
> > > 2. repeated sampling (×3 candidates) from gpt-oss-20B,
> > > 3. repeated sampling (×3 candidates) from Qwen3-Coder-30B,
> > > 4. an ensemble strategy where each model (DeepSeek-Coder-V2-16B, gpt-oss-20B, and Qwen3-Coder-30B) contributes one candidate (total of 3 candidates).
> > >
> > > For each strategy, we select the best candidate per source program and report functional correctness and execution time.
> > > To quantify diversity, we introduce **CSSim**[1], defined as the average pairwise distance between candidates based on AST edit distance. Higher CSSim indicates greater structural and edit-level diversity.
> > >
> > >
> > > |  | Functional Correctness (% ↑) | Execution Time (ms ↓) | Diversity (CSSim ↑) |
> > > |---| :---: | :---: | :---: |
> > > |  DeepSeek-Coder-V2-16B | 40.4  | 781  |  26.7 |
> > > |  gpt-oss-20B |  42.1 |  762 |  29.4  |
> > > |  Qwen3-Coder-30B | 43.6  |  753 |  30.7 |
> > > |  LLM Ensemble | **47.2**  | **734**  |  **33.8** |
> > >
> > >
> > > As shown in the table above, the ensemble strategy achieves the highest diversity (CSSim = 33.8), compared to repeated sampling from individual models (26.7–30.7).
> > > This confirms that cross-model ensembling introduces more diverse optimization strategies at the structural level.
> > > Moreover, increased diversity correlates with improved performance, with the ensemble achieving both higher functional correctness and lower execution time.
> > >
> > >
> > > ---
> > >
> > > We thank the reviewer for the helpful feedback. We will incorporate the suggested clarifications into the camera-ready version. We are happy to clarify further if needed.
> > >
> > >
> > > [1] Li H, Zhou X, Shen Z. Rewriting the code: A simple method for large language model augmented code search[C]//Proceedings of the 62nd Annual Meeting of the Association for Computational Linguistics (Volume 1: Long Papers). 2024: 1371-1389.

---

### Decision · Program_Chairs · 2026-04-30

**Decision:**

Accept (regular)

**Comment:**

This paper studies a problem in LLM-based code translation: balancing functional correctness with runtime efficiency. While existing methods primarily focus on correctness, they often neglect the execution efficiency of the translated code. To bridge this gap, the authors propose SWIFTTRANS, a framework that generates multiple candidate translations and selects the optimal one based on both correctness and efficiency.

The rebuttal was highly constructive and successfully addressed all the reviewers' primary concerns. Please ensure that all promised additions are incorporated into the final version.